# Improving Body Mass Index of School-Aged Children Using a Nine-Week Rope Skipping Training Intervention: A One-Group Pre-Test Post-Test Design

**DOI:** 10.3390/children9111715

**Published:** 2022-11-09

**Authors:** James Boadu Frimpong, Michael Agyei, Daniel Apaak, Edward Wilson Ansah, Larissa True

**Affiliations:** 1Department of Kinesiology, New Mexico State University, Las Cruces, NM 88011, USA; 2Department of Health, Physical Education and Recreation, University of Cape Coast, Cape Coast PMB TF0494, Ghana

**Keywords:** physical activity, motor skill development, weight status, youth

## Abstract

Studies examining the effectiveness of rope skipping training to improve the body mass index (BMI) of school children are scarce. Hence, this study examined the effectiveness of nine-week skipping training on the BMI of primary six school-aged children (*n* = 77). The participants underwent 30 min of skipping training three days per week over a nine-week period. The participants’ BMI was measured at the baseline and during weeks 3, 6 and 9. The results from a one-way ANOVA with repeated measures indicated a statistically significant difference in the BMI for both males [*F*(3,111) = 9.42, *p* < 0.001, *ηp*^2^ = 0.203] and females [*F*(3,114) = 7.35, *p* < 0.001, *ηp*^2^ = 0.162], suggesting an improvement in BMI. Post hoc comparisons with a Bonferroni adjustment revealed significant differences in BMI after nine weeks of intervention for males between the pre-test (*M* = 21.47, *SD* = 4.94) and the 9-week post-test (*M* = 20.15, *SD* = 4.36), and for females between the pre-test (*M* = 21.56, *SD* = 5.80) and the 9-week post-test (*M* = 20.68, *SD* = 5.32). This study demonstrated that regular participation in vigorous physical activity such as skipping training could promote child health by preventing the likelihood of young children being overweight. This result has implications for the inclusion of skipping training into the school life of school-aged children to help manage their BMI levels.

## 1. Introduction

Physical inactivity and childhood obesity are on the upsurge and thus have become critical public health concerns to nations across the world [1,2]. Globally, from 1975 to 2016, it is estimated that obesity prevalence increased in girls from 0.7% to 5.6% and 0.9% to 7.8% in boys [3]. This worldwide trend is alarming, especially given the parallels between obesity and physical inactivity. Mears [4] established that about 11% of children in the United States live a sedentary life, while 66% do not meet the recommended physical activity (PA) threshold. Another study revealed that 55.4% of 11–12-year-old school children in the US spend their entire day engaging in sedentary activities, while 41.7%, 2.2% and 0.7% spend their day doing light, moderate and vigorous PA, respectively [5]. School children in Ghana spend about eight hours in school, with a greater portion of the time being used for sedentary activities [6,7]. Considering this, physical education (PE) classes and recess breaks have been included on the school schedule to encourage pupils to be active and to reduce boredom [8,9,10,11]. Nonetheless, anecdotal experience suggests that some school children refuse to participate in PE classes due to the unenjoyable nature of such engagements. As this phenomenon persists, serious consequences come to bear on the physiological responses of these children, thus affecting their proper development and maintenance of their optimum health status [11,12,13].

Skipping, also known as jumping rope or rope jump, is a form of aerobic exercise that involves one or more participants skipping over a rope which is swung such that it passes under the feet and over the head of the participants in a continuous, uninterrupted manner [14]. Skipping may consist of only one person swinging and jumping over the rope or more than two participants, two of whom swing the rope while the other participants jump over the rope [15]. The rope is portable and relatively inexpensive compared to other exercise equipment [16]. Skipping has been performed by children as a fun game, used by boxers as a warm-up exercise and by regular exercisers as a cardio exercise since time immemorial [17,18]. Recently, many countries around the globe have incorporated skipping in their PE lessons [16] in an effort to help children achieve a better quality of life, and improve or maintain their physical fitness via PA. This was informed by the recommendation of Janssen and LeBlanc [19], who indicated that children and youth should accrue 30 min of daily moderate-to-vigorous intensity PA for optimum health.

Several researchers have reported the efficacy of skipping on various health outcomes. Jahromi and Gholami [20] established that jump rope training was an effective intervention for improving the physical fitness levels of 9–10-year-old female students. Grivedehi, Nourbakhsh and Sepasi [21] observed that skipping exercises have positive effects on all gross motor skills or the bodily fitness and movement factors, especially in school-age children. Additionally, positive effects of skipping have been found on the blood circulation in the heart, flexibility, balance, muscle strength, coordination, speed, vertical jumping and rhythm [22,23].

Some studies have shown that performing skipping for a period of six weeks can improve the physical fitness of the participants [24,25,26]. Other studies also reported a significant improvement in the physiological variables of participants after performing skipping for seven weeks [16,27], eight weeks [28,29], 12 weeks [30,31], 15 weeks [19] and 40 weeks [32]. Based on the successes of these skipping interventions to improve the participants’ physical fitness variables after a minimum of six weeks and a maximum of 40 weeks, it was conceptualized that nine weeks of skipping training may positively influence certain health outcomes of school-aged children. Up until now, it appears no recent study has tested the efficacy of a nine-week skipping intervention to improve the BMI of primary six school-aged children, creating a gap in the available literature. Moreover, the study focused on BMI because BMI, which is directly related to childhood obesity, has recently become a global health concern and it is also related to several adverse physical and psychosocial developmental constructs which could predispose children to eventual obesity during adulthood [33,34]. Hence, examining the efficacy of using a simple and relatively inexpensive training intervention such as rope skipping to improve the BMI of school children is worthwhile. Further, BMI is commonly used worldwide as a valid estimate of weight status due to its low cost and ease of administration [35,36]. Findings from this study could inform those who develop and implement school curricula to include skipping training in the PE curriculum and classes to improve the health of school children. Essentially, this study could also serve as an important reference for future studies in PA interventions that seek to reduce the increasing sedentary behaviors of school-aged children.

## 2. Materials and Methods

### 2.1. Participants and Procedures

This study used a one-group pre-test post-test repeated measures design. This design was used because the study examined the effectiveness of a skipping intervention conducted over time (i.e., nine weeks) [37,38] and it was not possible to randomly assign participants into distinct groups [39]. The study was conducted at University Primary School (UPS) in the Cape Coast Metropolis of Ghana. The population comprised all the primary six school-aged children of the UPS. Previous studies have described the population used in this study as school-aged children [9,10,11]. Out of a population of 240 primary six school children, 2 classes were randomly selected for recruitment (i.e., 40 school children in each class). Of the original sample of 80 school children, 77 participants completed the study. The three participants who dropped out were regularly absent from school which resulted in several missing data points; hence, they were excluded from the analyses.

Approval and permission to conduct the study was obtained from the Institutional Review Board (IRB) of a university in Ghana. Prior to the data collection, the research team sought approval from the school administrators and teachers and obtained signed consent forms from the parents of the participants. Afterwards, the researchers met with all the school-age children of the two randomly selected classes to explain the purpose and experimental protocols of the study. Data were collected by five trained research assistants who were recruited and trained on measurement protocol. The data collection lasted for nine weeks, with 30 min skipping training sessions occurring three times a week, on Mondays, Wednesdays and Fridays, during PE classes.

### 2.2. Measures

#### 2.2.1. Skipping Frequency

Skipping frequency was the number of jumps participants completed within the space of time given during the training sessions. The participants used the skipping rope to perform all the skipping activities. The skipping activity was performed after warm-up and stretching activities throughout the nine weeks. During the days of testing (i.e., the baseline, weeks 3, 6 and 9), however, the participants’ skipping frequency was the number of times they skipped within three minutes, and this was monitored and counted by the researchers and other trained research assistants.

#### 2.2.2. Height, Weight, BMI and BMI Percentiles

A locally manufactured stadiometer with a maximum calibration of 210 cm was used to measure the height of the participants in meters. The Camry electronic personal weighing scale was used to measure the weight of the participants in kilograms (kg). The participants’ BMIs were derived using the formula: BMI = weight/height ×; height in kg/m^2^. The BMI-for-age percentiles were calculated using the CDC growth chart interpretations: underweight (less than 5th percentile), healthy weight (5–85th percentile), at risk of overweight (85–95th percentile) and overweight (above 95th percentile) [40,41,42,43]. To determine whether a person is underweight, of a healthy weight, at risk of being overweight or overweight, the BMI scores (kg/m^2^) are plotted against the age (years) of the person according to their sex.

#### 2.2.3. Inclusion and Exclusion Criteria

All participants whose parents or guardians confirmed that they had no cardiovascular diseases, joint problems, were not under any form of medication and had the appropriate apparel (athletic wear and non-slip sports shoe with socks) were included in the study. Essentially, participants who did not report being sick at training sessions were allowed to participate. However, school children whose parents or guardians had confirmed that they were not fit for the study were dropped.

### 2.3. Data Analysis

Descriptive statistics (means and standard deviation) were conducted for the age, skipping frequency, weight, height and BMI for the male and female school children. The variables of the study were tested for normality, outliers, homogeneity of variance and linearity and no assumptions were violated. The BMI of the participants was measured at four different time points (i.e., pre-test, 3-week post-test, 6-week post-test and 9-week post-test). Moreover, since the BMI determination for males and females differ, the data were split before conducting the analysis and the results are presented separately for males and females. To determine if statistical differences in the BMI existed between the pre- and post-test measurements after nine weeks of skipping, a one-way analysis of variance (ANOVA) with repeated measures was employed. The decision criterion was set at a 95% confidence level. All the analyses were conducted using SPSS software version 21 (IBM Corp., Armonk, NY, USA).

## 3. Results

According to the CDC BMI-for-age growth charts, males (mean age = 11.92) and females (mean age = 11.72) fell in between the 25th and 50th percentiles on their respective charts, indicating a healthy or normal weight at the baseline. Table 1 shows the demographic and anthropometric data of the participants at the baseline.

The results from the one-way repeated measures ANOVA are in Table 2. The results indicated that there was a statistically significant difference in the pre-BMI and at least one of the post-BMI scores for both males [*F*(3,111) = 9.42, *p* < 0.001, *ηp*^2^ = 0.203] and females [*F*(3,114) = 7.35, *p* < 0.001, *ηp*^2^ = 0.162]. The partial eta squared indicated that for males and females, 20.3% and 16.2% of the decrease in their BMI, respectively, could be attributed to the skipping intervention. The post hoc comparisons employing the Bonferroni adjustment revealed that for male school children, no significant reductions in BMI were identified between the means of the pre-test (*M* = 21.47, *SD* = 4.94) and 3-week post-test (*M* = 21.43, *SD* = 4.81; *p* = 0.99, MD = 0.37). After six weeks of intervention, still no major differences were recorded between the pre-test (*M* = 21.47, *SD* = 4.94) and 6-week post-test (*M* = 21.31, *SD* = 4.63; *p* = 0.16, MD = 0.55). Significant differences were, however, indicated after nine weeks of intervention between the pre-test (*M* = 21.47, *SD* = 4.94) and 9-week post-test (*M* = 20.15, *SD* = 4.36; *p* = 0.01, MD = 1.32), signifying an improvement in the BMI status of the male participants between the beginning of the intervention and the conclusion. Essentially, a 6.1% percentage decrease in BMI was noted for the male participants following the nine-week skipping intervention.

For female school children, no substantial reductions in BMI were identified between the pre-test (*M* = 21.56, *SD* = 5.80) and 3-week post-test (*M* = 21.25, *SD* = 5.47; *p* = 0.82, MD = 0.31). After six weeks of intervention, there were still no substantial differences between the means of the pre-test (*M* = 21.56, *SD* = 5.80) and 6-week post-test (*M* = 21.07, *SD* = 5.41; *p* = 0.36, MD = 0.48). However, statistically significant differences were recorded after nine weeks of intervention between the pre-test (*M* = 21.56, *SD* = 5.80) and 9-week post-test (*M* = 20.68, *SD* = 5.32; *p* = 0.01, MD = 0.87), indicating an improvement in BMI. The BMI of the female participants was reduced by 4.1% following the nine-week skipping intervention.

## 4. Discussion

The current study examined the effect of a nine-week, thrice weekly rope skipping intervention on the BMI of primary six school children in Ghana. The study found that there was a statistically significant reduction in BMI between the baseline and the 9-week post-test scores for both the male and female school children. Our findings imply that the nine-week skipping training intervention program was effective in improving the BMI of primary six school-age children of the UPS, Cape Coast. These findings parallel results from other investigations [16,20,24,25,26,27,28,29,32].

For instance, Mullur and Jyoti [29] studied the impact of eight weeks of skipping training on the BMI of 12–16-year-old school children. The study revealed that the skipping training conducted over eight weeks significantly improved the BMI of the children in the experimental group, while children in the control group experienced no such improvements. The commonality between the current study and Mullur and Jyoti’s study is that skipping training over the course of several weeks improved the BMI in youths. However, the current study did not realize any substantial improvements until the nine-week mark, while Mullur and Jyoti recorded this same effect at eight weeks. This could be the result of differences in training intensity, or the mean age of the youths in Muller and Jyoti’s study being older and more physically mature, and thus intervention effects occurred in a shorter time period.

Similarly, Lee and In [26] investigated the effect of a skipping exercise program on the body composition of 12 female college students over a period of six weeks. The study yielded a statistically significant improvement in the BMI of the participants following the six weeks skipping exercise program. The finding that skipping training performed after some weeks improved the BMI as reported by Lee and In is similar with the current study. However, while Lee and In used only six weeks to note this effect, the current study used nine weeks. A possible reason could be because of the differences in the sample used. Using only females in Lee and In’s study might have accounted for an improvement in the BMI just after six weeks, which could be linked to females not having an improved fitness compared to males. It could also be that the participants in Lee and In’s study were not in the healthy weight zone, which probably led to an improvement in their BMI over a shorter time period. Additionally, age differences might have also played a role in our finding because, compared to college students who might have a relatively stable body composition at the baseline, school children on the other hand are still growing and maturing.

The findings of the current study contradict the findings of Chao-Chien and Yi-Chun [30], who studied the effect of a skipping rope intervention on the health-related physical fitness of intellectually impaired students aged 13–15. The study revealed no statistically significant improvement in the BMI of the participants after the training program. This, according to Chao-Chien and Yi-Chun, was because of not controlling the diet of the participants. It is important to note that the participants of the current study were not intellectually impaired and clearly had no challenges with rope skipping, whereas those in Chao-Chien and Yi-Chun’s study may not have benefited from the skipping rope intervention partially because of their intellectual impairment.

Although the current investigation used only one group, there was still a percentage decrease of 6.1% and 4.1% in the BMI (i.e., a marked improvement) of the male and female school children, respectively, after the nine-week skipping intervention program. This improvement could be attributed to the high intensity nature of the skipping training causing a high degree of the breakdown of accumulated fat in the body [44,45]. Another reason accounting for this finding could be the established positive link between a regular participation in vigorous PA and positive health outcomes, irrespective of body size and age [46,47]. The participants at their age were still growing, some very rapidly due to their growth spurts, and even though they were gaining weight and height over the course of the intervention, they still maintained and even improved their BMI. It should also be noted that the study took place soon after the long-term suspension of sporting activities in schools because of COVID-19, which rendered school-age children inactive for quite some time. This period of inactivity followed by the intervention may have played a role in the improvement of their BMI after the nine weeks of engaging in the skipping training intervention program [44,48,49].

## 5. Limitations

The study used only school-age children of a normal weight, which might have influenced the findings. Additionally, the use of only one group (i.e., without a control group) limits the generalizability of the study to all primary six school children. Essentially, attributing the improvement in the BMI of the participants to only the skipping training may be misleading. Future studies should consider using both the control and experimental groups to eliminate any possible extraneous factors that might have influenced the findings. Additionally, since BMI is highly related to height, it is possible that the height changes in the participants over the nine-week period might have influenced the findings, hence, future studies should adopt a more precise assessment of participants’ weight status. Lastly, although skipping is known to be a vigorous physical activity, the heart rate of the school children was not assessed, hence, future studies could assess the heart rate of the participants to be sure of the intensity.

## 6. Practical Implications

Given the surge in sedentary behaviors among school-aged children, the need for physical education teachers to make their classes more enjoyable and fun for school children has also increased. Skipping training, which requires just a simple rope, is relatively economical and enjoyable for school-age children and can be a good way of increasing physical activity and, subsequently, physical fitness in a relatively short period of time. Including skipping training in school activities can play a significant role in reducing the sedentary behaviors of school-age children and could possibly become a lifelong form of physical activity.

## 7. Conclusions

This study demonstrated that regular participation in vigorous PA, such as a skipping training intervention over a period of nine weeks, could act as an efficacious means of promoting child health by maintaining and even reducing BMI among school-age children. This result has implications for the inclusion of skipping training in the school life of school-age children to help manage their BMI levels.

## Figures and Tables

**Table 1 children-09-01715-t001:** Demographic and anthropometric data of participants at baseline.

Gender	Variable (Pre-Test)	*M*	*SD*
Male	Age (years)	11.92	0.59
	SF	149.32	64.73
	H (cm)	150.96	5.49
	W (kg)	49.02	12.16
	BMI (kg/m^2^)	21.47	4.94
Female	Age (years)	11.72	0.46
	SF	135.03	57.45
	H (cm)	153.91	7.99
	W (kg)	51.28	14.88
	BMI (kg/m^2^)	21.56	5.80

N (male = 38, female = 39). SF = skipping frequency; H = height; W = weight; BMI = body mass index.

**Table 2 children-09-01715-t002:** One-way repeated measures ANOVA for the effect of nine-week skipping on BMI.

Gender		*M*	*SD*	*F*	*df*	Sig.	Partial Eta Squared (*ηp*^2^)
				9.42	3,111	<0.001	0.203
Male (*n* = 38)	BMI Pre-test	21.47 *	4.94				
	BMI 3-week post-test	21.43	4.81				
	BMI 6-week post-test	21.31	4.63				
	BMI 9-week post-test	20.15 *	4.36				
				7.35	3,114	<0.001	0.162
Female (*n* = 39)	BMI Pre-test	21.56 **	5.80				
	BMI 3-week post-test	21.25	5.47				
	BMI 6-week post-test	21.07	5.41				
	BMI 9-week post-test	20.68 **	5.32				

Male: significant at *p* < 0.05 = * Female: significant at *p* < 0.05 = **.

## Data Availability

Data are available upon reasonable request from the corresponding author.

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
