# Peer review of "Improving Body Mass Index of School-Aged Children Using a Nine-Week Rope Skipping Training Intervention: A One-Group Pre-Test Post-Test Design"

_children, 2022, doi:10.3390/children9111715_

Round 1
Reviewer 1 Report
It is a good work but I have to suggest that the significates of this study are not enough for being publicized on an ISI journal. Because, as the information in the paper, it can be found the skipping has been revealed by former studies that a suitable sport item to improve the fitness and BMI of school children by former studies. I suggest giving the more convincing significates of this study.
Author Response
Point 1: It is a good work but I have to suggest that the significates of this study are not enough for being publicized on an ISI journal. Because, as the information in the paper, it can be found the skipping has been revealed by former studies that a suitable sport item to improve the fitness and BMI of school children by former studies. I suggest giving the more convincing significates of this study.
Response 1: This has been addressed in the final paragraph of the introduction (page 2, lines 75-86), the addition of the Practical Implication section (page 6, lines 250-258), and the changes that have been made per Reviewer 2’s suggestions. Thank you.
Reviewer 2 Report
Thank you for the opportunity to review this interesting manuscript. The study is interesting, actual and it provides useful information for a cheap and feasible intervention in children/adolescents. The Authors proposed a 9-weeks intervention of skipping rope as a school physical activity. Despite the topic is interesting, it presents some important limitation.
The first limitation is about the terms adopted to define the population. The Authors adopted children, pupils and other terms that could not be associated with people with a mean age of 11.72 / 11.92 years. Please, better define and justify with references the terms adopted for the population. Eventually, change “pupils” also in the title.
The second important limitation is the use of the BMI, it is a value strongly correlated with height and in 9 weeks the height of people aged 11 years could change and influence the results. Even if the Authors highlighted this point, I think that in limitation this topic should be deeply presented. Future studies should adopt more precise test to evaluate body fat percentage.
The last limitation is related to the level of the physical activity. The Authors had no collected data about the heart rate or RPE scales of the children and consequently they cannot conclude that this is a vigorous physical activity. Please, define better the level of the exercise
The Authors started the introduction with the first paragraph about physical activity level in the United States. After this first paragraph, the Authors started the second paragraph with a sentence about physical activity in Ghana school children. First, I strongly suggest to provide to the reader an overview of the world-wide physical activity level otherwise, if it is not possible, I suggest to merge the two paragraphs.
Line 36-42: there is no citation for these sentences, I strongly suggest to read and use one of the most recent publication on this topic “Are Physically Active Breaks in School-Aged Children Performed Outdoors? A Systematic Review”
Line85: the Authors correctly, highlighted that 3 participants didn’t completed the study. Is it possible the reasons of the dropout?
Did the study followed the Helsinki principles? Please, specify this.
Line 132: SPSS software, please provide manufacture information.
Line 134-135: M age: please write M as mean and not only M
Table 1: SF, H, W, BMI: please write their extended version in notes.
Please, add the clinical/practical applications of the study
Author Response
Point 1: The first limitation is about the terms adopted to define the population. The Authors adopted children, pupils and other terms that could not be associated with people with a mean age of 11.72 / 11.92 years. Please, better define and justify with references the terms adopted for the population. Eventually, change “pupils” also in the title.
Response 1: This has been addressed throughout the manuscript by changing “pupils” to “school-aged children.” The title has been changed to “Improving body mass index of school-aged children using a nine-week rope skipping training intervention: A one-group pre-test post-test design.” Thank you.
Point 2: The second important limitation is the use of the BMI, it is a value strongly correlated with height and in 9 weeks the height of people aged 11 years could change and influence the results. Even if the Authors highlighted this point, I think that in limitation this topic should be deeply presented. Future studies should adopt more precise test to evaluate body fat percentage.
Response 2: Thank you for this observation, and we agree that using BMI is a limitation particularly in this age group. This has been addressed in the last paragraph of the introduction (page 2, line 75-79) and has been expanded in the limitations (page 6, lines 244-257). We have also provided empirical support for the use of BMI in youth by adding two references (page 2, lines 81-82).
Point 3: The last limitation is related to the level of the physical activity. The Authors had no collected data about the heart rate or RPE scales of the children and consequently they cannot conclude that this is a vigorous physical activity. Please, define better the level of the exercise
Response 3: This has been addressed. Please see page 6, lines 247-249.
Point 4: The Authors started the introduction with the first paragraph about physical activity level in the United States. After this first paragraph, the Authors started the second paragraph with a sentence about physical activity in Ghana school children. First, I strongly suggest to provide to the reader an overview of the world-wide physical activity level otherwise, if it is not possible, I suggest to merge the two paragraphs.
Response 4: The first and second paragraphs have been merged, and these sentences have been added to emphasize the global increase in obesity and its relation to physical inactivity: “Globally, from 1975-2016, it is estimated that obesity prevalence increased in girls from 0.7% to 5.6% and 0.9% to 7.8% in boys [3]. This worldwide trend is alarming, especially given the parallels between obesity and physical inactivity.” See page 1, lines 29-32.
Point 5: Line 36-42: there is no citation for these sentences, I strongly suggest to read and use one of the most recent publication on this topic “Are Physically Active Breaks in School-Aged Children Performed Outdoors? A Systematic Review”
Response 5: This has been addressed. The material on the suggested topic above and others have been incorporated.
Point 6: Line85: the Authors correctly, highlighted that 3 participants didn’t completed the study. Is it possible the reasons of the dropout?
Response 6: This information has been provided; please see page 3, lines 98-100. Thanks for the observation.
Point 7: Did the study followed the Helsinki principles? Please, specify this.
Response 7: Yes, this study followed the declaration of Helsinki and the research protocol was approved by the Institutional Review Board of University of Cape Coast which is mandated to review all human subject research. This statement can be found in the manuscript under “Institutional Review Board Statement” (see page 7, lines 271-273).
Point 8: Line 132: SPSS software, please provide manufacture information.
Response 8: Manufacturer information has been provided on page 4, line 147.
Point 9: Line 134-135: M age: please write M as mean and not only M
Response 9: This has been addressed. Thanks for this observation. (See page 4 lines 149 and 150).
Point 10: Table 1: SF, H, W, BMI: please write their extended version in notes.
Response 10: This has been addressed.
Point 11: Please, add the clinical/practical applications of the study
Response 11: The practical implication of this study has been added (see page 6, lines 250-258).
Round 2
Reviewer 1 Report
NO more suggestions
Reviewer 2 Report
Thank you for the corrections and improvements made in the manuscript. It results improved and better presented.